# Financial Development and CO₂ Emissions in Post-Transition European Union Countries

**Yilmaz Bayar** [1], **Laura Diaconu (Maxim)** [2,*] **and Andrei Maxim** [3]

[1] Department of Economics, Faculty of Economics and Administrative Sciences, Usak University, Usak, Turkey; yilmaz.bayar@usak.edu.tr

[2] Department of Economics and International Relations, Faculty of Economics and Business Administration, "Alexandru Ioan Cuza" University of Iasi, Carol I Avenue, no. 22, Iasi, Romania

[3] Department of Marketing, Management and Business Administration, Faculty of Economics and Business Administration, "Alexandru Ioan Cuza" University of Iasi, Carol I Avenue, no. 22, Iasi, Romania; andrei.maxim@uaic.ro

**\*** Correspondence: laura.diaconu@feaa.uaic.ro or lauradiaconu_07@yahoo.com; Tel.: +40-232-201-399

**Abstract:** Carbon dioxide emissions are on the rise, posing a serious global issue. Therefore, it is important that policymakers identify the exact causes of these emissions. This paper investigates the influence of financial development, primary energy consumption, and economic growth on CO₂ emissions in 11 post-transition European economies. The assessment was made for the 1995–2017 period using panel cointegration and causality analyses. The causality analyses did not reveal significant connection between financial sector development and CO₂ emissions, but rather a two-way causality between primary energy consumption and economic growth, on one hand, and CO₂ emissions on the other. Meanwhile, long-run analysis disclosed that financial sector development and primary energy consumption positively affected CO₂ emissions. Our results seek to grab the attention of policy makers, who could work towards creating country-specific strategies that balance the relationship between financial development and CO₂ emissions. These long-term policies could ensure both development of the financial sector and environmental protection.

**Keywords:** financial development; energy consumption; economic growth; CO₂ emissions; panel data analysis

## 1. Introduction

Global climate change is the biggest environmental problem of the 21st century and a subject of debates among scientists, policymakers, and environmentalists. Sustainable development, in this context, has shifted people's perspective from short-term growth to long-term socio-economic and environmental development. Researchers have started to investigate the costs and factors of environmental degradation. A major factor that is known to cause climate change is the increased concentration of greenhouse gases, of which carbon dioxide (CO₂) accounts for more than 75% [1].

The literature regarding the relationship between economic growth and environmental degradation in general and CO₂ emissions and economic growth in particular includes a wide range of theoretical and empirical studies. Most research works studying the relationship between economic growth and CO₂ emissions started their analysis by using the Kuznets hypothesis, which states that as a country begins to develop, there is a positive relationship between economic growth and environmental degradation. This occurs because, in the initial developmental stages, countries tend to disregard environmental pollution in a bid to enhance economic growth. However, with increase in per capita income and welfare levels, environmental degradation significantly decreases,

while growth continues to rise. Therefore, an inverted-U-shaped relationship between economic growth and environmental degradation has been indicated by many studies [2,3].

The literature regarding the link between economic growth and $CO_2$ emissions is also consistent. Bruyn et al. [4] and Ozturk and Acaravci [5] reported an eventual positive relationship between economic growth and $CO_2$ emissions. However, considering the fact that economic growth is a complex process that leads to structural changes [6], it is also important to include in the analysis the factors that lead to economic progress. Therefore, while some researchers, such as Shahbaz, Hye, Tiwari, and Leitão [7] or Omri et al. [8], found a positive relationship between $CO_2$ emissions and trade openness, others indicated that $CO_2$ emissions are positively associated with urbanization rate [8,9] or development of the financial sector, since they have a larger share in a more developed economy [10,11].

Studies have also identified other factors that influence economic growth and indirectly increase or decrease $CO_2$ emissions. For example, Nag and Parikh [12] mentioned the income effect of economic growth, highlighting its role in increasing $CO_2$ emissions in India. Economic structures may also increase carbon emissions. For instance, changes in economic structures in both oil-producing and non-oil-producing sub-Saharan countries [13] and in China's export industry [14] have considerably increased $CO_2$ emissions. Zhang et al. [15] concluded that a higher level of economic growth and an increased share of the tertiary industry can significantly reduce $CO_2$ emissions, while a high level of urbanization has an adverse effect. Meanwhile, other researches have shown that the adjustments of energy and industry structures can decrease carbon intensity [16,17]. The same effect would have a higher end-use efficiency of the electricity industry [18].

Most of the studies that were conducted after 2000 and that focused on the relationship between economic growth and $CO_2$ emissions have also started to include energy consumption in their analyses. The results for various countries in Central America [19] and Asia (China [20,21], Malaysia [22], India [23], Japan [24], and Turkey [25]) concluded that non-renewable energy consumption increased $CO_2$ emissions. In order to reduce environmental pollution and energy dependency, several countries have moved towards renewable energy sources [26]. A study conducted by the International Energy Agency showed that in Organization for Economic Co-operation and Development (OECD) states, renewable energy consumption increased by 140.73% between 1974 and 2014 [27]. Meanwhile, at the global level, renewable energy consumption increased by an annual average of 2.6% between 2012 and 2014 [28]. In Turkey, Dilli and Nyman [29] found that, in 2015, renewable energy was the second-largest domestic energy source, after coal. However, the impact of renewable energy production on $CO_2$ emissions in this country is unclear. While Bolük and Mert [30] mentioned that per capita electricity production from renewable energy sources reduces per capita $CO_2$ emissions, Bulut [31] underlined a positive relationship between two variables—electricity production from renewable sources increases total $CO_2$ emissions.

This paper focuses on the influence of financial development, primary energy consumption, and economic growth on $CO_2$ emissions in 11 post-transition economies (Bulgaria, Croatia, Czechia, Estonia, Hungary, Latvia, Lithuania, Poland, Romania, Slovakia, and Slovenia) during 1995–2017. Our research will make a significant contribution to the literature because exhaustive empirical study of the impact of financial development on environmental quality in these economies has not been done yet. Another novel aspect of this research work is its methodological approach. The analysis involves the usage of panel cointegration and causality tests, robust to cross-sectional dependence and heterogeneity. The paper is structured as follows: In the next section, we present a brief literature overview of the theoretical and empirical approaches conducted on the topic. Section 3 underlines the data and the methods used, Section 4 discusses the results of our analysis, and Section 5 concludes the paper.

## 2. Literature Review

The relationship between environmental degradation and economic growth has been widely discussed in previous studies, with most of them indicating an "inverted-U-shaped" link between the two variables, as defined by the Environmental Kuznets Curve [32,33]. Economic growth has

typically been measured with the help of GDP or GDP per capita. One of the first analyses of the relationship between GDP per capita and the environmental pollution indicators belongs to Panayotou [34], whose conclusions empirically supported the Kuznets Curve. Before him, Shafik and Bandyopadhyay [35] also indicated a U-shaped relationship between environmental pollution and economic growth. Grossman and Krueger [36] stated that a rise in sulphur dioxide and dark-matter concentrations can be correlated with an increase in "per capita income". This association can be observed up to a certain level, after which these concentrations decrease, even though "per capita income" continues to grow. Latter, Selden, and Song [37] underlined that, as the level of economic growth increases, environmental pollution, measured through the $CO_2$ emissions, also rises.

In the beginning of the 21st century, the studies underlining an inverted-U-shaped link between $CO_2$ emissions and economic growth have significantly increased. For example, Ang [38] found this relationship to exist in the case of France, Jalil and Mahmud [20] in China, and Saboori et al. [39] in Malaysia. Meanwhile, Tsai [40] demonstrated the inverted-U-shaped curve in the relationship between GDP and environmental degradation on a panel of 62 economies. Other studies have included a smaller number of countries—six Central American economies [19] or 19 states [41], but their results have also confirmed the Environmental Kuznets Curve. However, this type of link between GDP and environmental degradation was not confirmed in the study conducted by Richmond and Kaufmann [42] on 36 countries, or in the research of Halicioglu [43] or Ozturk and Acaravci [5] on Turkey. Moreover, other studies indicated a linear causal connection between GDP and $CO_2$ emissions [44] or a bidirectional causality linking the two variables in the cases of Turkey [45] and Tunisia [46]. Meanwhile, Friedl and Getzner [47] indicated an N-shaped relationship between GDP per capita and $CO_2$ emissions.

Recent studies have included other variables in their analyses of economic growth and $CO_2$ emissions. Some of them indicated that some financial variables could reduce $CO_2$ emissions in emerging economies only when greater degrees of liberalization and financial sector development are achieved [6]. Similar results were found in the case of China: Between 1997 and 2011, the level of financial development was correlated with a reduction of $CO_2$ emissions in the developed regions and an increase of emissions in the less-developed ones [48]. Overall, financial development has received an increased attention in both empirical and theoretical analyses regarding environmental degradation.

From a theoretical standpoint, the impact of financial development on $CO_2$ emissions can be approached from four perspectives [49]. The first one refers to the situation in which the financial markets offer domestic companies the resources necessary to acquire environmentally friendly technologies for manufacturing [50]. The second approach refers to the fact that financial development may increase $CO_2$ emissions when it involves foreign investment inflows. This could be explained by the fact that these investments will increase the amount of energy used. However, contradictory results were found by Paramati et al. [51] on a panel of the G20 countries. Their conclusions underlined that foreign direct investment (FDI) significantly reduced $CO_2$ emissions in both developing and developed economies. Yuxiang and Chen [52] also showed that development of the financial sector in China, determined by the presence of foreign investors, encouraged the usage of advanced and green technologies, which, in turn, reduced $CO_2$ emissions. Thirdly, financial development might increase the number and scale of manufacturing activities, which would lead to an increase in land degradation, pollution, and carbon emissions [49]. The last approach refers to the fact that financial development injects more money into the economy, which will increase the consumption and, consequently, the energy demand for producing goods and services [48].

The empirical approaches regarding the relationship between financial development and $CO_2$ emissions have also included some other economic and institutional variables in the analysis. Tamazian et al. [53], who analyzed the influence of financial development on environmental degradation in Brazil, Russia, India, China, the United States, and Japan, concluded that, apart from the financial sector, the GDP, trade liberalization, and institutions have also had an important role in determining $CO_2$ emissions. A more comprehensive study, performed on 40 European countries,

was conducted between 1985 and 2014 by Sy et al. [54] in order to investigate the relationship between financial development, carbon emissions, economic growth, and trade openness. They indicated the presence of a neutral relationship between financial development and carbon emissions in the analyzed states.

Alom et al. [55] investigated the link between carbon emissions, urbanization, financial development, and energy consumption in Bangladesh between 1985 and 2015. They showed that financial development can have a positive impact on carbon emissions. The urbanization process was also included in the analyses conducted by Destek and Ozsoy [56], Ozatac et al. [57], and Pata [25], in which the Environmental Kuznets Curve was empirically tested.

Sadorsky [58], investigating the effect of urbanization on $CO_2$ emissions in several emerging economies, concluded that the impact of urbanization is positive, but insignificant. Using the Granger causality test, Hossain [9] noticed that, in the long run, there is no significant causal association between $CO_2$ emissions, economic growth, trade openness, energy consumption, and urbanization in newly industrialized countries. However, in the short term, some unidirectional relations were identified; for example, from economic growth to $CO_2$ emissions. However, Wang et al. [59], using the same statistical method—the Granger causality test—obtained different results. They showed that, in the short run, urbanization positively influences both energy consumption and $CO_2$ emissions. Meanwhile, in the long term, urbanization, combined with energy consumption, will lead to higher $CO_2$ emissions. Interesting conclusions were drawn by Wang et al. [60] on OECD states. Apart from the fact that they found an inverted-U-shaped curve between urbanization and $CO_2$ emissions, their conclusions also showed that massive energy consumption, together with economic growth, will increase per capita $CO_2$ emissions. Contrary results were found by Charfeddine and Ben Khediri [61]. They investigated the case of UAE states during the period 1975–2011, and underlined two important conclusions. The first one indicated an inverted-U-shaped curve between financial development and $CO_2$ emissions, and the second finding highlighted that electricity consumption, urbanization, and trade openness improve environmental quality. A study conducted by Nasreen et al. [62] on South Asian countries during the period 1980–2012 suggests that, while financial stability improves environmental quality, a higher economic growth, energy consumption, and population density have a negative impact on the environment in the long run.

Energy consumption has also been a largely debated aspect by both economists and environmentalists who analyzed the relationship between economic and financial development, on one hand, and $CO_2$ emissions, on the other hand. Trying to achieve higher growth rates through industrialization, the developing states have increased their consumption of oil and fossil fuels, which, in turn, have significantly augmented $CO_2$ emissions. However, depletion of natural resources, combined with water, air and, soil pollution determined policy makers to search for alternative energy sources [63]. Researches that include renewable energy production and consumption in the analysis of the Environmental Kuznets Curve are relatively new. Sulaiman et al. [64] noticed that, in the case of Malaysia, per capita electricity production from renewable energy sources reduced per capita $CO_2$ emissions between 1980 and 2009. Similar results were found by Lopez-Menendez, Perez, and Moreno [65] for the 27 EU states. However, even if per capita renewable energy sources diminished per capita $CO_2$ emissions in all the analyzed countries between 1996 and 2010, the U-shaped curve relation between economic growth and environmental degradation was confirmed only in the cases of Cyprus, Greece, Slovenia, and Spain. The positive impact of producing and consuming renewable energy on reducing $CO_2$ emissions was also empirically proven in 27 developed states for the period 1990–2012 [66]. This study, together with that conducted by Farhani and Shahbaz [67] on 10 Middle East and North African states during the period 1980–2009, also confirmed the validity of the Kuznets Curve hypothesis. However, Farhani and Shahbaz [67] argued that per capita consumption of renewable energy increased per capita $CO_2$ emissions in the analyzed countries. Other studies showed that per capita consumption of electricity from renewable sources did not have any effect on per capita $CO_2$ emissions and, moreover, the Kuznets Curve hypothesis could not be validated [68].

## 3. Data and Method

As mentioned before, the present study investigates the influence of financial development, primary energy consumption, and economic growth on $CO_2$ emissions in 11 post-transition economies during the period 1995–2017, with the help of panel cointegration and causality tests. While the dependent variable was proxied by carbon dioxide emissions, the financial sector development was represented by the financial development index of the International Monetary Fund (IMF) [69]. The economic impact of financial development has been an extensively investigated topic across theoretical and empirical literature. In these studies, financial development is generally proxied by domestic credits offered to the private sector, stock market capitalization, and the M1, M2, and M3 monetary aggregates. However, we preferred the financial development index of the IMF [69] because, unlike the other indicators of financial sector development, it simultaneously considers the financial markets' depth, the access to finance, and the efficiency of the financial markets [70,71]. While energy consumption was represented by primary energy consumption, economic growth was expressed through the evolution of the real GDP. The data regarding the $CO_2$ emissions and the primary energy consumption were obtained from BP [72], while the financial development index was taken from the IMF [69] database and the real GDP from the World Bank database [73]. All of the data were annual (see Table 1).

**Table 1.** Dataset definition.

| Variable | Abbreviation | Data source |
|---|---|---|
| Carbon dioxide emissions (million tons of carbon dioxide) | $CO_2$ | BP (2019) |
| Financial development index | FINDEV | IMF (2019) |
| Primary energy consumption (million tons oil equivalent) | ENERGY | BP (2019) |
| GDP (constant 2010 USD) (million dollars) | RGDP | World Bank (2019) |

The following empirical model was conceived in order to analyze the effect of the financial sector development, energy consumption, and economic growth on the $CO_2$ emissions in a country i (i = 1, …, 11) in year t (t = 1995,.., 2017).

$$CO_{2it} = f(FINDEV_{it}, ENERGY_{it}, RGDP_{it}) \tag{1}$$

The sample of the econometric analysis comprised the following countries: Bulgaria, Croatia, Czech Republic, Estonia, Hungary, Latvia, Lithuania, Poland, Romania, Slovakia, and Slovenia. The empirical analysis was conducted with the help of Stata 14.0, Eviews 10, and Gauss software. The key features of the dataset are presented in Table 2 and Table 3. It can be noticed that, on one side, the average $CO_2$ emissions amounted to approximately 66.8 million tons of carbon dioxide, and the average primary energy consumption was about 24.3 million tons oil equivalent. On the other side, the average financial development index was about 0.34 and the average real GDP was about 106.24 billion USD.

**Table 2.** Descriptive statistics of the dataset.

| | $CO_2$ | FINDEV | ENER | RGDP |
|---|---|---|---|---|
| Mean | 66.794 | 0.343 | 24.280 | 106,246.3 |
| Median | 37.200 | 0.342 | 17.500 | 55,033.04 |
| Maximum | 354.300 | 0.575 | 103.400 | 601,720.6 |
| Minimum | 6.900 | 0.107 | 3.200 | 10,351.09 |
| Std. Dev. | 85.569 | 0.098 | 25.662 | 116,618.2 |
| Skewness | 2.163 | −0.013 | 1.817 | 2.194 |
| Kurtosis | 6.696 | 2.768 | 5.543 | 7.872 |

**Table 3.** Relationships between variables.

|        | FINDEV | ENER   | RGDP  |
|--------|--------|--------|-------|
| FINDEV | 1.0000 | 0.1703 | 0.290 |
| ENER   |        | 1.0000 | 0.442 |
| RGDP   |        |        | 1.000 |

The panel causality and cointegration tests were used to see the short- and long-run impacts of the financial development, primary energy consumption, and economic growth on the $CO_2$ emissions. Westerlund and Edgerton's LM bootstrap cointegration test [74] was preferred, considering the sample size and the existence of cross-sectional dependence among the series—the LM bootstrap cointegration test allows autocorrelation and heteroscedasticity and offers better results in the case of smaller samples. Furthermore, Dumitrescu and Hurlin's causality test [75], a modified version of the traditional Granger causality test, was used to see the interaction between variables due to the presence of heterogeneity and cross-sectional dependence.

The long-run effect of the financial sector development, primary energy consumption, and economic growth on the $CO_2$ emissions was analyzed with the help of Westerlund and Edgerton's LM bootstrap cointegration test [74]. The test considers the dependence both within and between the individual cross-section units and it allows autocorrelation to differ among the cross sections. The test also investigates the joint null hypothesis of cointegration for all of the cross-sections, unlike the analysis of Pedroni [76,77] and of Banerjee and Carrion-i-Silvestre [78]. Westerlund and Edgerton's LM bootstrap cointegration test [74] is based on the Lagrange multiplier test of McCoskey and Kao [79]. Furthermore, the bootstrap cointegration test relies on sieve sampling and it considerably reduces the distortions of the asymptotic test.

We assume the following panel data model:

$$y_{it} = \alpha_i + x'_{it} + \beta_i + Z_{it},$$
$$Z_{it} = u_{it} + v_{it} \ \text{ and } \ v_{it} = \sum_{j=1}^{t} \eta_{ij}. \tag{2}$$

$$w_{it} = \sum_{j=0}^{\infty} \alpha_{ij} e_{it-j} \tag{3}$$

The hypothesis is tested through the following LM model, where the cross-sectional dependence is non-existent:

$$LM_{NT^2}^+ = \sum_{i=1}^{N} \sum_{t=1}^{T} \widehat{w}_{it}^{-2} S_{it} \ . \tag{4}$$

$S_{it}$ is a part of full modified estimation $Z_{it}$, while $\widehat{w}_{it}^{-2}$ is the estimation of $u_{it}$ (long-run variance).

The LM bootstrap cointegration test yields biased results in case of the existence of a cross-sectional dependence. It also shows when asymptotically standard normal distribution is very susceptible to serial correlation. Therefore, the bootstrap approach is used instead of a standard normal distribution to overcome the problem.

The long-run coefficients were estimated using the DSUR (Dynamic Seemingly Unrelated Cointegrating Regressions) estimator, developed by Mark et al. [80], which accounts for both heterogeneity and cross-sectional dependence. The endogeneity problem is also eliminated through the lags and leads included in the model. Lastly, the causality interactions among the series were analyzed with the help of the causality test of Dumitrescu and Hurlin [75]. The test is the modified version of Granger's causality test [81] regarding heterogeneity. It also yields reliable results in the case of small samples and the presence of cross-sectional dependence. The null hypothesis, known as HNC (Homogenous Non-Causality), posits no significant causality for any cross-sections of the panel. In the test, the null hypothesis is tested for each cross-section and, then, the panel $W_{N,T}^{HNC}$ statistic is calculated by averaging N standard Wald statistics $(W_{i,T})$ [79].

$$W_{N,T}^{HNC} = \frac{1}{N} \sum_{i=1}^{N} W_{i,T} \tag{5}$$

The panel-standardized $Z_{N,T}^{HNC}$ statistic is calculated by using $W_{N,T}^{HNC}$:

$$Z_{N,T}^{HNC} = \sqrt{\frac{N}{2K}} \left( W_{N,T}^{HNC} - K \right) \rightarrow N(0,1) \tag{6}$$

## 4. Results and Discussions

The specification of cross-sectional dependence and heterogeneity in the dataset is important for using the most reliable and correct unit root and cointegration tests. As a consequence, cross-sectional dependence was checked with the help of Breusch and Pagan's LM test [82], Pesaran's LM CD test [83], and the $LM_{adj.}$ test of Pesaran et al. [84]. The tests' results are presented in Table 4.

All of the tests disclosed the presence of a cross-sectional dependence between the $CO_2$ emissions, on one hand, and the financial sector development, primary energy consumption, and economic growth, on the other hand. The unit root and cointegration analyses will be explored through the second-generation tests.

**Table 4.** Cross-sectional dependence tests' results.

| Test | Test statistic | p-value |
|------|---------------|---------|
| LM | 45.851 | 0.001 |
| LM adj | 40.938 | 0.000 |
| LM CD | 40.114 | 0.000 |

In the second sub-stage of the pre-tests, the homogeneity of the cointegration coefficients was analyzed by using the homogeneity tests of Pesaran and Yamagata [85]. The test results are shown in Table 5. The null hypothesis in favor of homogeneity was rejected. Thus, the cointegration coefficients were revealed to be heterogeneous.

**Table 5.** Homogeneity tests' results.

| Test | Test statistic | p-value |
|------|---------------|---------|
| $\tilde{\Delta}$ | 9.563 | 0.000 |
| $\tilde{\Delta}_{adj.}$ | 9.224 | 0.000 |

The first-generation panel unit root test postulates that all of the cross-sections are independent and that they are equally affected by any shock of the cross-sections. However, the cross-sectional dependence tests disclosed the presence of cross-sectional dependence among the series. Therefore, the unit root in the variables was checked with the CIPS (Cross-Sectional IPS) [86] unit root test of Pesaran [87] by considering the presence of a cross-sectional dependence. The test results are displayed in Table 6. The results revealed that the variables *CO2*, *FINDEV*, *ENERGY*, and *RGDP* were I(1).

**Table 6.** Panel CIPS (Cross-Sectional IPS) unit root test's results

| Variables | Level | | First differences | |
|-----------|-------|---|-------------------|---|
| | **Constant** | **Constant + Trend** | **Constant** | **Constant + Trend** |
| $CO_2$ | −0.953 | −1.045 | −8.325* | −9.106* |
| FINDEV | −1.067 | −1.113 | −9.331* | −10.043* |
| ENERGY | −0.852 | −9.885 | −9.679* | −10.118* |

| RGDP | −0.906 | −1.146 | −10.362* | −10.977* |

* It is significant at 5% significance level.

The long-run interaction between the $CO_2$ emissions, on one hand, and the financial sector development, primary energy consumption, and economic growth, on the other hand, was investigated by the Westerlund and Edgerton's LM bootstrap cointegration test [74], taking into account the presence of a cross-sectional dependence and heterogeneity. The test results could be seen in Table 7. The test findings disclosed a cointegration relationship when the structural breaks were taken into account. So, the series moved together in the long run.

**Table 7.** Westerlund and Edgerton (2007) LM Bootstrap cointegration test results.

| | Constant | | | Constant + Trend | | |
|---|---|---|---|---|---|---|
| | Test statistic | Asymptotic p-value | Bootstrap p-value | Test statistic | Asymptotic p-value | Bootstrap p-value |
| $LM_{N^+}$ | 1.062 | 0.261 | 0.393 | 1.382 | 0.431 | 0.505 |

Note: Bootstrap probability values were derived from 10.000 repetitions, while asymptotic probability values were obtained from standard normal distribution. Lag and lead values were taken as 2.

Since the cointegration test disclosed that the series are cointegrated, we proceeded to the next step in the empirical analysis and we estimated the cointegration coefficients with the DSUR estimator, used by Mark et al. [80], by taking into account the cross-sectional dependence and the heterogeneity. The estimation results are presented in Table 8.

The long-run test estimations disclosed that financial development and primary energy consumption positively impacted $CO_2$ emissions, while real GDP negatively affected $CO_2$ emissions at the level of the entire panel. However, the financial sector development positively influenced the $CO_2$ emissions in Estonia, Hungary, Poland, Romania, Slovak Republic, and Slovenia at the country level. Meanwhile, the primary energy consumption positively affected the $CO_2$ emissions in Bulgaria, Croatia, Estonia, Latvia, Lithuania, Poland, Romania, Slovak Republic, and Slovenia. Lastly, the economic growth negatively influenced the $CO_2$ emissions in Bulgaria, Croatia, Czech Republic, Estonia, Hungary, Poland, Romania, Slovak Republic, and Slovenia.

Our results are consistent with those obtained in other theoretical and empirical studies. For instance, Dasgupta et al. [88] mentioned that an increase in the energy consumption, determined by financial development, may augment the $CO_2$ emissions. An empirical investigation conducted on India revealed that, in the long run, financial development increases environmental degradation [89]. In Pakistan, a one-percent increase in the financial development augments the $CO_2$ emissions by 0.165% [90]. Similar findings were obtained in the case of China: The lower the level of financial development, the more reduced the carbon emissions were between 1997 and 2013 [91]. Despite these results that are consistent with our findings, Ayeche et al. [92], using a panel data analysis, empirically proved a neutral relationship between financial development and carbon emissions in the European states between 1985 and 2014. Meanwhile, Al-Mulali, Ozturk, and Lean [93] found, with the help of a panel-pooled FMOLS model, that financial development could increase carbon emissions in the long run in the European states. Sadorsky [94] has also noticed that financial development and energy consumption lead to an increase in environmental degradation in the Central and Eastern European states. However, the negative impact on the environment could be reduced if the financial development determines a more efficient usage of energy by using more advanced technologies and if the increased funds enhance regulations regarding environmental protection [53]. These aspects have been empirically confirmed by Al-Mulali, Tang, and Ozturk [95], who used dynamic OLS and Granger causality tests and concluded that financial development can improve the environmental quality in both the short and long run. Indeed, if we consider the financial development that occurs after the "per capita income" has reached the benchmark point from the Environmental Kuznets Curve, when the economic growth increases while environmental degradation decreases, it is obvious that the high level of financial resources would enhance the

usage of advanced technologies. However, in the case of our study, the results showed a contradictory aspect: Even if the analyzed states seem to have surpassed the point on the Kuznets Curve after which economic growth is negatively correlated with environmental degradation, the financial development still increases pollution. Perhaps, in the case of our sample, the explanations could be found in the institutional factors because, as noticed by Tamazian and Bhaskara [11], the institutional framework could control the impact of financial development on carbon emissions.

**Table 8**. Long-run coefficients' estimation.

| Countries | FINDEV | ENERGY | RGDP |
|---|---|---|---|
| Bulgaria | 0.056 | 1.642* | –0.184* |
| Croatia | 0.031 | 1.117* | –0.105* |
| Czech Republic | 0.092 | 0.904 | –0.563* |
| Estonia | 0.105* | 0.483* | –0.197* |
| Hungary | 0.003* | 0.392 | –0.295* |
| Latvia | 0.067 | 0.493* | –0.541 |
| Lithuania | 0.018 | 0.417* | –0.230 |
| Poland | 0.083* | 0.356* | –0.185* |
| Romania | 0.209* | 0.475* | –0.381* |
| Slovak Republic | 0.472* | 0.409* | –0.651* |
| Slovenia | 0.398* | 0.229* | –0.217* |
| Panel | 0.183* | 0.495* | –0.372* |

* It is significant at 5% significance level.

Note: The problems of autocorrelation and heteroscedasticity were eliminated through the Newey–West method.

Furthermore, the causality interaction between the cointegrating series was tested with the help of Dumitrescu and Hurlin's causality test [75], and the results are presented in Table 9. The causality analysis's results reveal a two-way causality between primary energy consumption and $CO_2$ emissions, as well as between economic growth and $CO_2$ emissions. The causality analysis disclosed no significant interaction between the financial sector development and the $CO_2$ emissions in the short run. However, a mutual interaction between primary energy consumption and economic growth, on one hand, and the $CO_2$ emissions, on the other hand, could be noticed in the short run.

**Table 9.** Dumitrescu and Hurlin (2012) causality test's results.

| Null hypothesis | Test | Test statistics | P-value |
|---|---|---|---|
| FINDEV↛ $CO_2$ | Whnc | 1.529 | 0.372 |
|  | Zhnc | 1.263 | 0.358 |
|  | Ztild | 1.778 | 0.325 |
| $CO_2$ ↛FINDEV | Whnc | 1.183 | 0.197 |
|  | Zhnc | 1.117 | 0.185 |
|  | Ztild | 1.089 | 0.173 |
| $CO_2$↛ENERGY | Whnc | 2.728 | 0.000 |
|  | Zhnc | 2.047 | 0.000 |
|  | Ztild | 2.382 | 0.003 |
| ENERGY ↛$CO_2$ | Whnc | 3.273 | 0.000 |
|  | Zhnc | 2.945 | 0.000 |
|  | Ztild | 3.072 | 0.007 |
| $CO_2$↛RGDP | Whnc | 3.226 | 0.000 |
|  | Zhnc | 3.275 | 0.000 |
|  | Ztild | 2.998 | 0.000 |
| RGDP↛$CO_2$ | Whnc | 3.263 | 0.013 |

| | | |
|---|---|---|
| Zhnc | 3.881 | 0.000 |
| Ztild | 3.907 | 0.000 |

## 5. Conclusions and Policy Implications

The ongoing increase in greenhouse gas emissions, mainly resulting from $CO_2$ emissions, has determined both policy-makers and researchers to investigate its causes and to search for solutions to diminish the resulting negative effects on environment, biodiversity, and ecosystem. In this study, we explored the short- and long-run effects of financial sector development, together with economic growth and primary energy consumption, on the $CO_2$ emissions in 11 post-communist EU countries. Even though various studies were conducted on this topic for countries in Asia, Africa and Western Europe, the studies performed on Central and Eastern European economies are very limited. Therefore, the findings of our study could have a positive impact both on the literature and on the future decisions of policy makers.

In our research, the short-run interactions among the financial sector development, economic growth, primary energy consumption, and $CO_2$ emissions were analyzed with the help of Dumitrescu and Hurlin's causality test [80]. The causality analysis disclosed no significant interaction between the financial sector development and the $CO_2$ emissions in the short run, but a mutual interaction between the primary energy consumption and economic growth, on one hand, and the $CO_2$ emissions, on the other hand, in the short run. The long-run interaction among the variables was also explored with the help of the Westerlund and Edgerton LM Bootstrap cointegration test [74]. The long-run test estimations disclosed that financial development and primary energy consumption positively influenced $CO_2$ emissions, while real GDP negatively affected $CO_2$ emissions, at the level of the entire panel. Meanwhile, the financial sector development had a positive impact on the $CO_2$ emissions in Estonia, Hungary, Poland, Romania, Slovak Republic, and Slovenia at the country level. Moreover, while primary energy consumption positively affected $CO_2$ emissions in Bulgaria, Croatia, Estonia, Latvia, Lithuania, Poland, Romania, Slovak Republic, and Slovenia, the economic growth negatively influenced the $CO_2$ emissions in Bulgaria, Croatia, Czech Republic, Estonia, Hungary, Poland, Romania, Slovak Republic, and Slovenia.

Our results are similar to the findings obtained in other studies. However, in the case of our research, the results showed a contradictory aspect: Even if economic growth is negatively correlated with environmental degradation in the case of the 11 analyzed states, financial development still leads to an increase in pollution. As shown in the literature, a well-functioning financial sector is vital for reducing the emissions of $CO_2$, because it may enhance technological innovations and environmentally friendly production processes. Therefore, in the case of our sample, a proper institutional framework, which should enact environmental protection regulations, could be a solution for reducing carbon emissions and ensuring a cleaner environment.

Our findings have several major policy implications. Despite the fact that other studies found that financial development could reduce $CO_2$ emissions in various states, our research indicates the opposite. In the case of the 11 post-transition economies that we have analyzed, the positive impact of the financial development on the $CO_2$ emissions may suggest that companies tend to expand their production though credit rather than to develop energy-saving technologies. This aspect should raise concerns with the policy makers concerning the environmental effects of financial development. They should balance the relationship between financial development and $CO_2$ emissions, according to the specific context of each country, and formulate long-term strategies for supporting both the financial sector and environmental protection. Therefore, governments should be focused more on allocating resources for nurturing technological progress in the industrial sector, such as providing loans for investments that generate products with lower carbon emissions and financing renewable resource projects. Such an approach could improve energy efficiency and, consequently, reduce carbon emissions. Other actions that could be taken involve providing support for the development of energy generation from renewable sources—hydro, solar, and wind—or allocating subsidies for adopting "green" technologies. Meanwhile, governments should focus on those strategies that blend economic incentives with regulatory measures aimed at alleviating $CO_2$

emissions. A first step would be to create those capacities required for reliable information collection and analysis, so that national institutions could accurately estimate emissions and make forecasts under alternative mitigation scenarios. The next step, necessary for the successful implementation of the mitigation strategies, would involve coordination among different public and private actors, such as various government ministries, research institutes, universities, and industries. Some of the most efficient strategies could be to establish a carbon tax, together with tradable carbon quotas.

Meanwhile, the financial institutions should take the initiative in protecting the environment. For example, they can offer loans with low interest rates to those who can deploy energy-efficient technologies.

Starting from the results obtained in this paper, we intend to expand the research beyond these 11 states to the rest of the European economies by including some other variables into the analysis. Therefore, in a future study, we intend to investigate the impact of the financial development, energy consumption, and economic growth on the quality of the environment in all of the European Union (EU) member states, grouped by their year of adhesion to the EU, in order to see the differences between and within these groups. An important aspect of these comparative analyses would be to highlight, for each state, the share of renewable energy sources in the primary energy consumption and the types of the conventional energy sources. Meanwhile, in order to have a more comprehensive image of the impact of GDP on the quality of the environment, we intend to conduct a detailed analysis regarding the branches of each EU economy that affect the size of the GDP.

**Author Contributions:** The authors contributed equally to this paper. Laura Diaconu (Maxim) and Andrei Maxim described the context of the analysis and conducted the literature review. Yilmaz Bayar developed the research approach and performed the data analysis. Yilmaz Bayar, with the help of Laura Diaconu (Maxim) and Andrei Maxim, described the main results. Laura Diaconu (Maxim) and Yilmaz Bayar formulated the main conclusions. Andrei Maxim and Laura Diaconu (Maxim) edited the paper.

**Funding:** "This research is funded by the Ministry of Research and Innovation within Program 1—Development of the national RD system, Subprogram 1.2—Institutional Performance—RDI excellence funding projects, Contract no. 34PFE/19.10.2018."

**Conflicts of Interest:** "The authors declare no conflict of interest."

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
