# Peer review of "Financial Development and CO2 Emissions in Post-Transition European Union Countries"

_sustainability, doi:10.3390/su12072640_

Round 1

Reviewer 1 Report

Dear Authors

The presented paper discusses the influence of financial development, primary energy consumption and economic growth on CO2 emissions in 11 post-transition European economies, over the period 1995-2017, through panel cointegration and causality analyses.

An approach to the issue the impact of financial development on CO2 emission is very interesting, but I have some concern.

The assumption related to determining the impact of financial development on the quality of the environment in the above mentioned economies was not met.

The conclusions section repeats the results of tests conducted without drawing conclusions for e.g. politicians or financiers. Post-communist countries are very diverse and identify specific areas of the economy or the financial mechanisms that affect the increase in CO2 emissions would be very valuable achievement in this work.

In my opinion, it would also be interesting to:

  • discussing the impact of national membership of the European Union by extracting data from the year of accession to the EU;
  • providing the share of renewable energy sources in primary energy consumption;
  • indication of the type of conventional energy sources;
  • specification of the branches of the economy affecting the volume of GDP.

Text editing

Minor editorial comments are marked in the manuscript.

Best regards

Reviewer

Author Response

Dear Reviewer,

We do thank you very much for your time and effort to revise our paper.

We have acknowledged all your valuable suggestions and we have made the appropriate corrections in the text. All the revisions have been made with Track Changes.

Please find, in the attached document, the detailed corrections we have made, according to your comments.

Sincerely yours,

The authors

Reviewer 2 Report

Dear Editors, dear authors,

I liked the idea and the manuscript in general, although there are some points which are preventing me at the moment from recommending it for publication.

While i do not see major issues with the work as such, i think the presentation has to be improved. Here are a few points i would like to see addressed:

  1. While i will refrain from specific comments about the English language since i am not a native English speaker, the language has to be improved. There are many errors in grammar, which make it hard to follow the manuscript. I would highly recommend a thorough proofreading by a native speaker before submitting a revised version of the manuscript.
  2. While there is an extensive introduction and literature review, i failed to grasp from those section why the methods which are later on presented were used.
  3. The conclusions and policy implications section does not really go beyond the presented work. I would like to see how one could expand on it, what are the next steps? Why do the authors think that the data differs to some other studies? How reliable are the results?

Author Response

(The authors gave the same response as above.)

Round 2

Reviewer 1 Report

Dear Authors

I am fully satisfied with the changes introduced so far and additions to the manuscript. The effort made to introduce changes influenced the higher quality of article.

I have only one doubt.

Please correct information in Table 3 (digits).

Best regards

Reviewer
